

# Experimental trials of species-specific bat flight responses to an ultrasonic deterrent

Sarah Rebecah Fritts[1], Emma Elizabeth Guest[1,2], Sara P. Weaver[2], Amanda Marie Hale[3,4], Brogan Page Morton[5] and Cris Daniel Hein[6]

[1] Department of Biology, Texas State University, San Marcos, Texas, United States
[2] Bowman, San Marcos, Texas, United States
[3] Department of Biology, Texas Christian University, Fort Worth, Texas, United States
[4] Western EcoSystems Technology, Inc., Cheyenne, Wyoming, United States
[5] Wildlife Imaging Systems, Hinesburg, Vermont, United States
[6] National Renewable Energy Laboratory, Arvada, Colorado, United States

Corresponding author
Sarah Rebecah Fritts,
fritts.sarah@txstate.edu

## ABSTRACT

Unintended consequences of increasing wind energy production include bat mortalities from wind turbine blade strikes. Ultrasonic deterrents (UDs) have been developed to reduce bat mortalities at wind turbines. Our goal was to experimentally assess the species-specific effectiveness of three emission treatments from the UD developed by NRG Systems. We conducted trials in a flight cage measuring approximately 60 m × 10 m × 4.4 m (length × width × height) from July 2020 to May 2021 in San Marcos, Texas, USA. A single UD was placed at either end of the flight cage, and we randomly selected one for each night of field trials. Trials focused on a red bat species group (*Lasiurus borealis* and *Lasiurus blossevillii*; $n = 46$) and four species: cave myotis (*Myotis velifer*; $n = 57$), Brazilian free-tailed bats (*Tadarida brasiliensis*; $n = 73$), evening bats (*Nycteceius humeralis*; $n = 53$), and tricolored bats (*Perimyotis subflavus*; $n = 17$). The trials occurred during three treatment emissions: low (emissions from subarrays at 20, 26, and 32 kHz), high (emissions from subarrays at 38, 44, and 50 kHz), and combined (all six emission frequencies). We placed one wild-captured bat into the flight cage for each trial, which consisted of an acclimation period, a control period with the UD powered off, and the three emission treatments (order randomly selected), each interspersed with a control period. We tracked bat flight using four thermal cameras placed outside the flight cage. We quantified the effectiveness of each treatment by comparing the distances each bat flew from the UD during each treatment *vs.* the control period using quantile regression. Additionally, we conducted an exploratory analysis of differences between sex and season and sex within season using analysis of variance. Broadly, UDs were effective at altering the bats' flight paths as they flew farther from the UD during treatments than during controls; however, results varied by species, sex, season, and sex within season. For the red bat group, bats flew farther from the UD during all treatments than during the control period at all percentiles ($p < 0.001$), and treatments were comparable in effectiveness. For cave myotis, all percentile distances were farther from the UD during each of the treatments than during the control, except the 90th percentile distance during high, and low was most effective. For evening bats and Brazilian free-tailed bats, results were inconsistent, but high and low were most effective, respectively. For tricolored bats, combined and low were significant at the 10th–75th percentiles, high was significant at all percentiles, and

combined was most effective. Results suggest UDs may be an effective means of reducing bat mortalities due to wind turbine blade strikes. We recommend that continued research on UDs focus on low emission treatments, which have decreased sound attenuation and demonstrated effectiveness across the bat species evaluated in this study.

## INTRODUCTION

Wind energy is rapidly increasing throughout the world in an effort to reduce use of fossil fuels, largely as a climate change mitigation measure; however, wind energy has the unintended consequence of bat mortalities resulting from wind turbine blade strikes (*Allison et al., 2019*). Collision mortalities have been documented at wind energy facilities worldwide (*Rydell et al., 2010*; *Arnett & Baerwald, 2013*; *Arnett et al., 2016*; *Zimmerling & Francis, 2016*; *Agudelo et al., 2021*). These impacts are of concern because of their potential population-level effects on certain bat species (*Frick et al., 2017*; *Friedenberg & Frick, 2021*) that have relatively low reproductive rates compared to other mammals of similar size (*Barclay et al., 2003*). Moreover, several species impacted by wind turbines also suffer from other natural and anthropogenic stressors, including white-nose syndrome, pesticides, and land-use changes (*Erickson et al., 2016*; *O'Shea et al., 2016*; *Frick, Kingston & Flanders, 2020*). With these synergistic effects threatening bat populations, there is high concern among regulators, conservationists, researchers, wildlife managers, and private industry about the risk wind turbines pose to bats. Although bat mortalities have been documented at wind energy facilities for about 40 years, there are limited minimization strategies that can be widely implemented, and the need for technological solutions continues to grow (*Hein & Hale, 2019*; *Friedenberg & Frick, 2021*). Long-distance migratory bat species are of particular concern in North America as mortalities of these species occur across the continent (*Arnett et al., 2005*; *Zimmerling & Francis, 2016*; *Choi, Wittig & Kluever, 2020*; *American Wind Wildlife Institute, 2021*), and it has been projected that at least one species, the hoary bat (*Aeorestes [Lasiurus] cinereus*), could experience population declines by up to 90% from wind energy alone (*Frick et al., 2017*; *Friedenberg & Frick, 2021*). Thus, these species often are targets of impact minimization strategies, such as curtailment (*Adams, Gulka & Williams, 2021*; *Whitby, Schirmacher & Frick, 2021*), deterrents (*Arnett et al., 2013*; *Romano et al., 2019*; *Weaver et al., 2020*), or a combination of the two (*Good et al., 2022*).

Investigations related to the influence of weather on bat mortalities at wind turbines reported significantly greater mortalities during nights with low wind speeds (*Arnett et al., 2005*; *Baerwald et al., 2009*; *Arnett et al., 2013*); thus, feathering turbine blades during periods of low wind speed was suggested as a viable curtailment strategy, also known as blanket curtailment. Blanket curtailment has been documented to reduce total bat

mortalities by 54–69%, hoary bat mortalities by 24–64%, eastern red bat (*L. borealis*) mortalities by 42–74%, and silver-haired bat (*Lasionycteris noctivagans*) mortalities by 30–66% (*Whitby, Schirmacher & Frick, 2021*). Another meta-analysis suggested a 63% decrease in total bat mortalities during operational minimization (*Adams, Gulka & Williams, 2021*). Nonetheless, this minimization strategy results in a loss of annual energy production that may not be financially sustainable for some wind energy facilities. To reduce the loss in annual energy production, curtailment strategies have advanced to incorporate threat prediction models using additional weather variables (*e.g.*, temperature) and acoustic bat activity (*Martin et al., 2017*; *Hayes et al., 2019*; *Peterson et al., 2021*; *Rabie et al., 2022*).

Ultrasonic deterrents (UDs) provide an alternative approach to curtailment and attempt to create a disruptive airspace to prevent bats from entering the rotor-swept area of a wind turbine. Echolocating bats supplement their spatial perception by emitting ultrasound and perceive their surroundings by listening to the reflected echoes (*Griffin, 1958*). This sense allows bats to orient, capture prey, communicate, and avoid obstacles in complete darkness. Bat's perception of ultrasound echoes can be diminished, or masked, by biological noises (*e.g.*, "clicks") emitted by moths (*Hristov & Conner, 2005*; *Corcoran et al., 2011*), it was hypothesized that broadcasting high-frequency transmissions from wind turbines may create a disorienting airspace, thus "jamming" a bat's ability to perceive its own echoes (*Szewczak & Arnett, 2007*). Acoustic deterrence has been demonstrated to decrease bat activity, foraging behavior, and flight performance, potentially due to auditory masking that precludes the use of echolocation (*Gilmour et al., 2021*).

Various UD technologies have been studied at wind energy facilities, with results varying among species at a given location or within species across different locations and times (*Arnett et al., 2013*; *Romano et al., 2019*; *Weaver et al., 2020*). The reasons for species-specific variability in effectiveness are unknown, but it may be related to variation in species' echolocation characteristics, ultrasound attenuation (*Arnett et al., 2013*; *Weaver et al., 2020*), and deterrent configuration (*Romano et al., 2019*). For example, a UD developed by NRG Systems deployed at a wind farm in Texas (USA) reduced bat mortalities for hoary bats and Brazilian free-tailed bats (*Tadarida brasiliensis*) by 78% and 54%, respectively, but no reductions in mortalities for other species in the genus *Lasiurus* were observed (*Weaver et al., 2020*). The GE Renewable Energy UD tested in Illinois (USA) reduced overall bat mortalities by 29%, but annual deterrent effectiveness varied for eastern red and silver-haired bats (*Romano et al., 2019*). In these studies, the observational data required to answer why differences exist among species is lacking. Improving the effectiveness of UDs across a wider range of species requires more controlled testing that allows for observations of individual bat flight paths and echolocation responses to various ultrasound configurations (*Romano et al., 2019*).

Our objective was to use a controlled study, including from those that echolocate at both high and low frequencies, to the NRG Systems UD (hereafter UD) using a large outdoor flight cage. We examined flight responses of five species of bats to various UD signals and hypothesized that deterrent signals with low-frequency sound would have a greater effect on low-frequency echolocating bats (*i.e.*, those with characteristic frequency <35 kHz),

whereas deterrent signals with high-frequency sound would have a greater effect on high-frequency echolocating bats (*i.e.*, those with characteristic frequency ≥35 kHz). In addition, we used an exploratory analysis to examine potential differences within each species between sex, season (fall *vs.* spring), and the interactive effects of sex and season. Understanding why and how bats interact with wind turbines continues to be an active area of research (*e.g.*, *Richardson et al., 2021*; *Guest et al., 2022*), and we hypothesized that responses of bats to UD signals could vary based on internal motivational states related to the timing of reproduction or migration.

## MATERIALS AND METHODS

To test the responses of individual, wild-captured bats of known species to the UD, we conducted a study at the Freeman Center, a 1,400-ha property owned by Texas State University in Hays County, Texas (29.9390, −98.0097 WGS 84) during fall 2020 (July–October) and spring 2021 (March–May). To meet objectives, we constructed an open-air flight cage specifically designed to test bat responses to UDs (Fig. 1). The open-air flight cage was approximately 60 m × 10 m × 4.4 m (length × width × height), was surrounded by 6.4-mm, lightweight, plastic netting (Industrial Netting, Minneapolis, Minnesota, USA), and had a UD mounted on a pole ~1.5 m high on each end. The dimensions of the flight cage were selected based on the precise goals of this project and the requirements for maintaining local bat species in enclosures. Height and width were based on insectivorous bat care standards from *Bat World Sanctuary (2010)* and recommendations from staff at Austin Bat Refuge (Austin, Texas, USA). The length was designed to accommodate the blade length of most modern land-based wind turbines produced prior to late 2019 (Fig. 1).

The two UDs used in this study consisted of a waterproof box with six subarrays (Fig. 1). Each subarray emitted a continuous sound at one of the following predetermined frequencies: 20, 26, 32, 38, 44, and 50 kHz. This frequency range encompasses the characteristic frequencies of most bat species known to occur in the United States and Canada and was the configuration of the NRG Systems UD at the time of testing. NRG Systems programmed the UDs to emit three treatments: low (emissions from subarrays at 20, 26, and 32 kHz), high (emissions from subarrays at 38, 44, and 50 kHz), and combined (emission from all six subarrays) and reported the UD to have a signal density of 125 dB at 1 m from the source.

We randomly selected one of the two UDs for each night of field trials. The UDs were powered by a generator positioned approximately 10 m from the flight cage and shielded by plywood boards to reduce noise. We mounted four AXIS 1942-e thermal video cameras (Axis 1942-e, Axis Communications, Lund, Sweden; hereafter "cameras") on the north side of the flight cage to limit nearby city heat signatures from interfering with bat thermal visibility (Fig. 2). We placed the cameras ~23 m from the cage at a height of 3.7 m so that the fields of view encompassed the entire flight cage and slightly overlapped between neighboring cameras. We programmed the cameras to record at 30 frames per second. We time-synched and monitored cameras using a cable-connected laptop at an observer

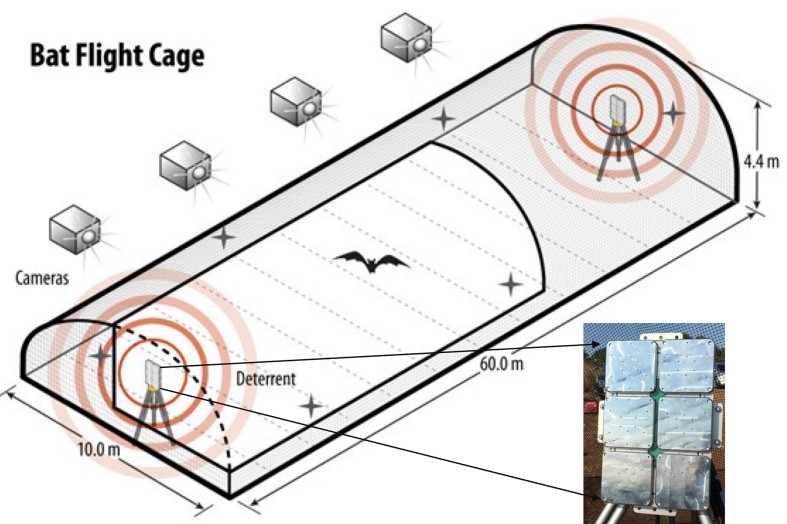

**Figure 1 The flight cage used to assess species-specific responses of bats to three ultrasonic deterrent emissions.** We assessed bat flight behavior to three ultrasonic deterrent emissions in a flight cage measuring (l × w × h) 60 m × 10 m × 4.4 m using four thermal cameras with overlapping fields of view. One ultrasonic deterrent was placed at each end of the flight cage and randomly selected each night for trials. The flight cage is on Texas State University property in San Marcos, Texas, USA.

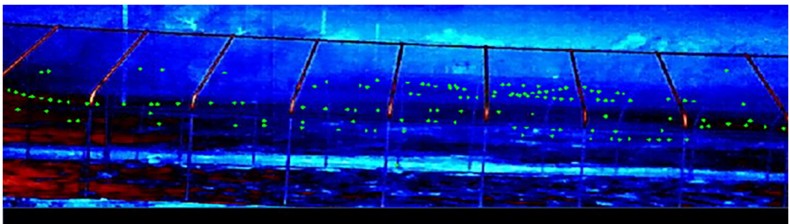

**Figure 2 Example output from one thermal camera showing bat locations during each video frame that we used to calculate distance from the ultrasonic deterrent.** We used bat location during each thermal video frame to calculate the various percentile distances that bats flew from the ultrasonic deterrent during the three emission treatments *vs.* a control period with the deterrent turned off.

station positioned 8 m from the end of the cage with the operating UD. During trials, we minimized observer sources of light, sound, and other potential causes of disruption.

We captured bats within 120 km of the flight cage (typically within 30 min of the flight cage in Hays County, Texas, USA) on both public and private properties for which we had authorization using mist nets, harp traps, and hand captures from July 13, 2020–October 7, 2020 for the fall season (45 trial days) and March 5–May 15, 2021 (40 trial days) for the spring season. We placed captured bats in cloth bags and placed cloth bags in 19-L buckets to transport in climate-controlled field vehicles. Because we held bats for several hours at a time, we fed bats meal worms (*Tenebrio molitor*) *ad libitum*. Once transported to the flight cage, we recorded the species and sex of each bat and released one bat into the flight cage at a time. A bat trial was 27 min and consisted of seven 4-min periods: an acclimation period

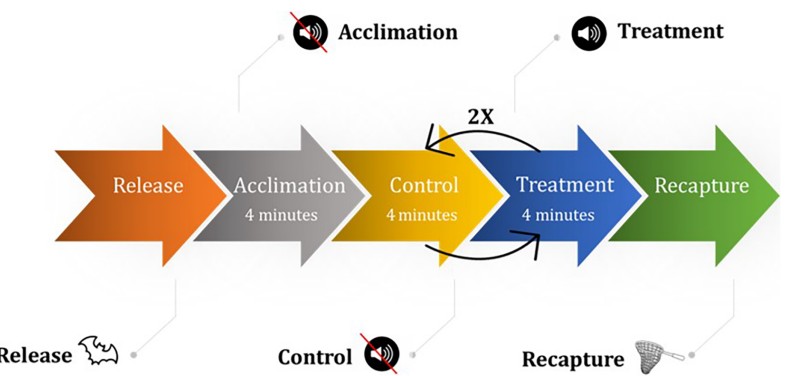

**Figure 3** The experimental design consisted of releasing a bat into the flight cage followed by an acclimation period then a control period then the three emission treatments, order selected randomly, each interspersed by a control. All periods lasted 4 min.

followed by a control period, then the three aforementioned UD emission treatments randomly ordered and each interspersed with a control period (Fig. 3). We only subjected bats to experimental trials if they flew during the acclimation period. We did not conduct trails during rain events. If at any point during the trial the bat stopped flying, we noted the occurrence and continued the trial. We omitted from analyses bats that used ≤50% of the flight cage during the entire trial to limit potential bias of a bat to a particular side of the flight cage for a reason unrelated to the study (*e.g.*, the side the bat was released on, an influence outside of the flight cage such as free-flying bats, the side the researchers were on). We typically held bats between 2–6 h and never overnight. The project was conducted under both a Texas Parks and Wildlife Department state permit (SPR-1217-243), which included protocols specific to the SARS-CoV-2 virus, and a Texas State University Institutional Animal Care and Use Committee permit (#6224). Additionally, we followed the National White-nose Syndrome Decontamination Protocol Version 10.14.2020 (*U.S. Fish and Wildlife Service, 2020*).

In 2021, we observed variation in pelage coloration among the red bats that we had presumed were eastern red bats based on known locality data. We speculated that some of these individuals were western red bats; therefore, we used the methods of *Korstian et al. (2015)* to confirm species identification for a subset of these bats (*n* = 12) using DNA extracted from fecal pellets that had been collected during the study. All of the sampled bats were western red bats (*L. blossevillii*) (unpublished data). Because our capture sites were in the known range of eastern red bats (*L. borealis*) and we did not confirm species identification for all captured red bats, we analyzed eastern and western red bats as one species group (*i.e.*, red bats).

We analyzed the resulting thermal videos using Python and the OpenCV library. We read each video frame (recorded at 30 frames/s) and applied a background subtractor to detect the movement of bats on the stationary background. We logged the coordinate (pixel location) and other feature information associated with each detection into a data frame. To eliminate erroneous detections due to noise in the video, we created a custom

 

filter that recorded only detections that had a nearby detection, based on Euclidean distance between coordinates in adjacent frames (before or after). This allowed us to retain the detections that were part of a continuous track while omitting those detections that had no spatial or temporal neighbors nearby. Because four cameras were used to cover the full length of the flight cage, the detections from all videos were aggregated into a single data frame and corrected for distortion. To aggregate, we used the $x$-pixel coordinate for each detection to estimate the distance from the operating UD. We generated the distance estimate using a per-camera calibration relating $x$-pixel coordinates to known distances within the flight cage. Once the conversion to a global coordinate system was complete, we consolidated the detection from each camera. To eliminate duplicate detections between cameras, we acquired the minimum distance values for each camera and restricted the adjacent camera detections to prevent detections from exceeding the next camera's minimum.

To assess differences in distance that each bat flew from the UD between the first control period and each UD emission treatment, we used quantile regression (*Cade, 2017*) in R (*R Core Team, 2021*) package (*Quantreg, 2022*). We used distance at each video frame that the bat was from the operating UD as the response variable; thus, during each 4-min period within the trial there were ~7,200 distance points for each bat, which are autocorrelated due to the small time between frames. We then used unique bat identification number and treatment (either control or one of the frequency emission treatments) as categorical independent variables and included the interactive term between unique bat identification number and treatment. This method is similar to using bat identification number as a random effect. Also, this modeling approach was more efficient than using a random effect and yielded similar results because the sample size for each bat is large. With this approach, we estimated all quantiles (tau = 1:99), but focused on the 10th, 25th, 50th, 75th, and 90th quantiles in subsequent analyses. Additionally, estimating quantiles of differences in distances allowed the ability to detect differences when they did not occur homogeneously across distances, which would be missed by mean regression models.

We conducted separate models for each treatment *vs*. control comparison due to the large sample size of distances obtained per 4-min period. The goal was to compare the quantile differences between each emission treatment and control period; however, the field trials included three distinct control periods. Therefore, we first assessed differences among control periods using an analysis of variance (ANOVA). We found no statistical difference among control periods; thus, we selected the first control period following the acclimation period for the pairwise comparisons to minimize differences in sample sizes that could bias results. We used a Bonferroni correction when comparing results from multiple treatments.

Although the previous analysis was the main objective, we also conducted *post-hoc* exploratory analyses to assess the influence of sex, season, and sex within season to the distance bats flew from the UD. When overall differences in distances between control and treatment periods were significant, we then assessed differences in season (spring *vs*. fall), sex, and the interaction between sex and season using the difference in flight distance from

the UD between the UD emission treatment and control period at each percentile as the response variable using separate ANOVAs. We could not add sex, season, or sex within season to the quantile regressions as the models would not converge.

## RESULTS

We conducted successful trials and analyses on 46 red bats, 57 cave myotis, 73 Brazilian free-tailed bats, 53 evening bats, and 17 tricolored bats. We omitted 3, 10, 4, 6, and 0 bats from those species, respectively, for not flying during trials or having a flight path that used <50% of the flight cage during the entire 24-min trial period. Responses differed by species, but a pattern emerged that indicated greater differences in distance at 0.10 to 0.50 quantiles with less difference at greater distances at 0.75 and 0.90 quantiles (Fig. 4).

For the red bat group, we conducted trials on fall males $n = 17$, spring males $n = 4$, fall females $n = 13$, spring females $n = 12$ (Fig. 5). Red bats flew farther from the UD during all treatments than during the control period at the 10–90th percentiles ($p < 0.001$), and treatments were comparable in effectiveness as estimated by the difference in flight distance between the treatments and the control periods (Tables 1 and 2). There was a significant season effect during the combined and high treatments and an interaction between sex and season during the low treatment; (Tables 3 and 4, Fig. 5). The difference in distance between control and treatment periods was greater in spring than in fall during combined and high treatments. For the low treatment, the greatest distance between control and treatment was for spring males, followed by spring females, and then all fall bats combined (Fig. 5).

For cave myotis, we conducted trials on fall males $n = 21$, spring males $n = 11$, fall females $n = 25$, spring females $n = 0$ (Fig. 6). Cave myotis flew further from the UD during each treatment than controls at all percentile distances except the 90th percentile distance during the high treatment (Tables 1 and 2). For this species, the low treatment was most effective as estimated by the difference in flight distance between the treatments and the control periods (Tables 1 and 2). Because we did not capture females during the spring, we did not include the interaction between sex and season in the ANOVAs for this species. The differences between control and treatment flight distance from the UD for cave myotis males were 3.174 m ($p = 0.019$) and 3.176 m ($p = 0.015$), greater than for females during the low treatment at the 75th and 90th percentiles, respectively (Tables 3 and 4, Fig. 6).

For evening bats, we conducted trials on fall males $n = 21$, spring males $n = 1$, fall females $n = 23$, spring females $n = 8$ (Fig. 7). The results were inconsistent for the combined and low treatments, and individuals of this species flew farthest from the UD compared to the control during the high treatment (Tables 1 and 2). We did not include the interaction between sex and season in the ANOVA due to the low sample size of spring males. There were no differences in the response by sex or season for this species (Table 3, Fig. 7).

For Brazilian free-tailed bats, we conducted trials on fall males $n = 31$, spring males $n = 7$, fall females $n = 21$, spring females $n = 14$ (Fig. 8). Results were inconsistent and variability was high, but the combined treatment was most effective as estimated by the difference in flight distance between the treatments and the control periods (Tables 1 and
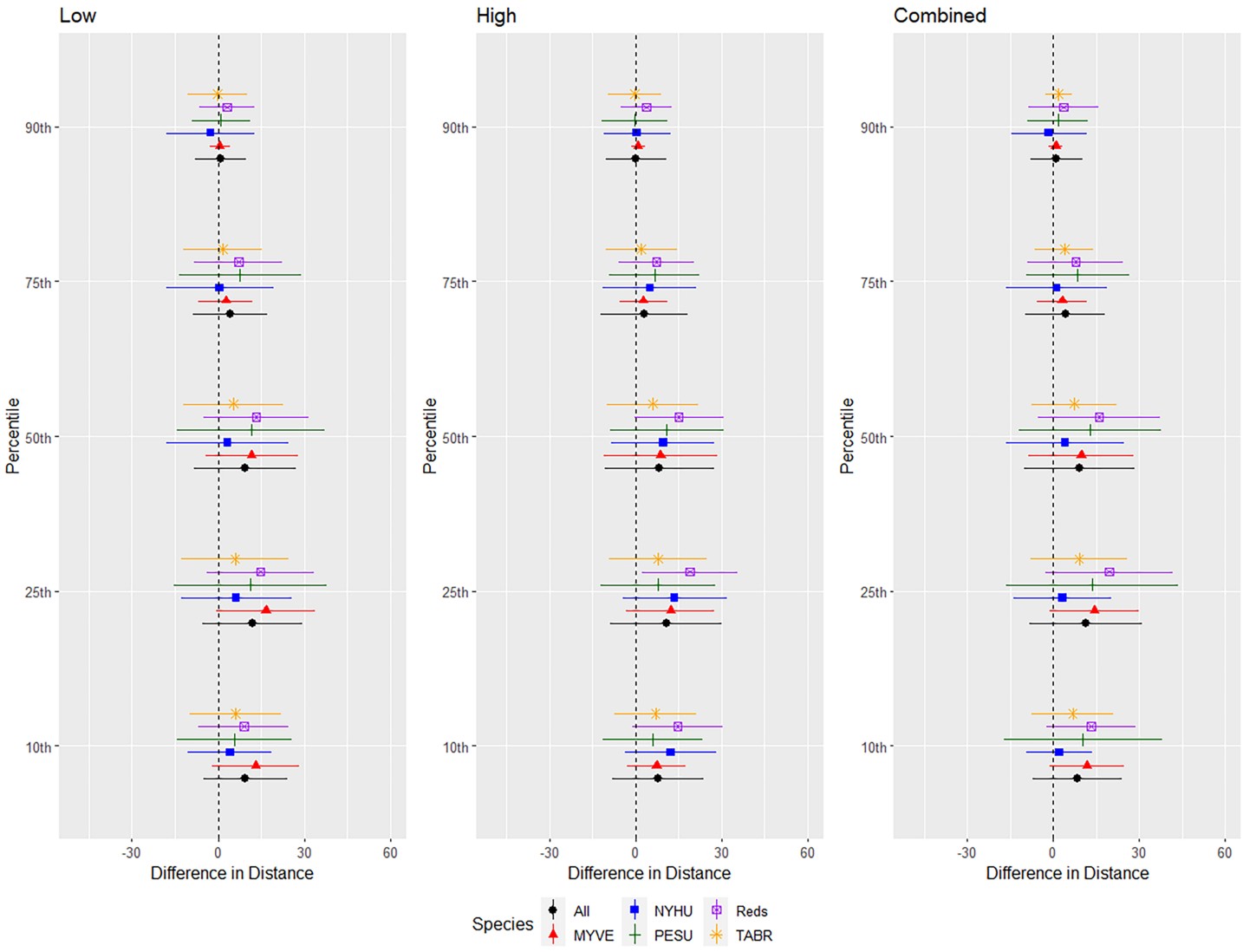

**Figure 4 Differences in distances bats flew during three ultrasonic deterrent emission treatments *vs.* controls with the ultrasonic deterrent turned off.** The differences in distance (m) that bats by species flew from the ultrasonic deterrent (UD) during each emission treatment (Combined, Low, High) *vs.* the control period with the UD powered off by sex, season, and sex within season. All = all bats, NYHU = Nycticeius humeralis Reds = Lasiurus borealis and Lasiurus blossevillii, MYVE = Myotis velifer, PESU = Perimyotis subflavus, TABR = Tadarida brasiliensis).

2, Fig. 8). Although the low treatment was significant at the 10, 50, 75, and 90th percentiles (Table 1), the difference in distances between treatment and control were low, and bats flew closer to the UD at the 90th percentile (Table 2). There were some differences in distance flown by sex and an interaction between sex and season for some percentiles; however, no discernable pattern was observed for this species (Fig. 8).

For tricolored bats, we conducted trials on fall males $n = 7$, spring males $n = 4$, fall females $n = 5$, spring females $n = 1$ (Fig. 9). The combined and low treatments were significantly different from the control at the 10–75th percentiles, whereas the high treatment was significantly different from the control at all percentiles (Tables 1 and 2). The combined treatment was most effective as estimated by the difference in flight distance

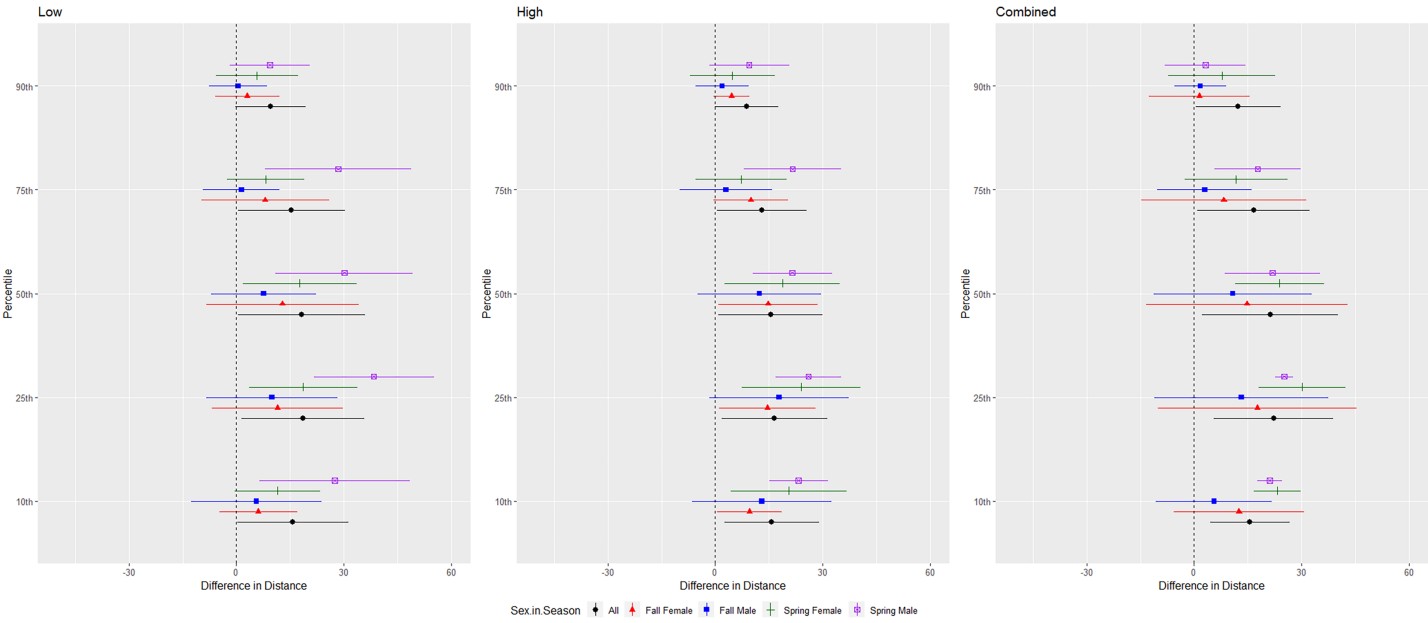

**Figure 5 Differences in flight distances that red bats (*Lasiurus borealis* and *Lasiurus blossevillii*) flew during three ultrasonic deterrent emissions *vs.* a control period.** The differences in distance (m) that red bats (*Lasiurus borealis* and *Lasiurus blossevillii*) flew from the ultrasonic deterrent (UD) during each emission treatment (Combined, Low, High) *vs.* the control period with the UD powered off by sex, season, and sex within season.

between the treatments and the control periods (Tables 1 and 2). We did not include the interaction between sex and season due to the low sample size of spring females. Tricolored bats consistently flew farther from the UD compared to the control in fall than spring during the low and high treatments at all percentiles and during the combined treatment at the 25th percentile (Tables 3 and 4, Fig. 9).

# DISCUSSION

Overall, the difference in distances flown between the UD emissions and control periods indicates that the ultrasonic broadcast signals tested were successful in shifting flight patterns away from the UD for four bat species and one bat species group in experimental trials in a flight cage environment. Nonetheless, the apparent effectiveness of the three treatments differed among species and in some cases in ways that were not clearly interpretable. These findings are consistent with previous research that assessed effectiveness of UDs at reducing bat mortality at operational wind turbines. Previous research studies reported species-specific differences in fatality reductions at wind turbines when UDs were emitting ultrasound compared to when the devices were turned off or at wind turbines without UDs (*Arnett et al., 2013*; *Romano et al., 2019*; *Schirmacher, 2020*; *Weaver et al., 2020*). In our study, Brazilian free-tailed bats flew farthest from the UD during the combined treatment. This finding is consistent with the results of *Weaver et al. (2020)* in which researchers tested the same UD device and frequency emissions as this study and reported a 54.5% reduction in Brazilian free-tailed bat mortality at operational wind turbines.

| Table 1 Results from quantile regression. | | | | | | |
|---|---|---|---|---|---|---|
| **Species** | **Treatment** | **Percentile** | **Beta** | **SE** | **t-value** | **p** |
| Red bats | Combined | 10th | 13.21 | 0.22 | −60.36 | <0.001 |
| | | 25th | 19.57 | 0.19 | −102.7 | <0.001 |
| | | 50th | 16.07 | 0.21 | −76.11 | <0.001 |
| | | 75th | 7.81 | 0.19 | −41.62 | <0.001 |
| | | 90th | −7.3 | 1.16 | 6.3 | <0.001 |
| | High | 10th | 14.59 | 0.2 | 72.7 | <0.001 |
| | | 25th | 18.83 | 0.21 | 88.03 | <0.001 |
| | | 50th | 15.11 | 0.23 | 65.87 | <0.001 |
| | | 75th | 7.32 | 0.19 | 38.42 | <0.001 |
| | | 90th | −12.16 | 1.14 | −10.63 | <0.001 |
| | Low | 10th | 9.01 | 0.25 | 35.69 | <0.001 |
| | | 25th | 14.82 | 0.2 | 73.38 | <0.001 |
| | | 50th | 13.29 | 0.6 | 22.23 | <0.001 |
| | | 75th | 7.12 | 0.19 | 37.31 | <0.001 |
| | | 90th | −8.33 | 1.37 | −6.1 | <0.001 |
| Cave myotis | Combined | 10th | 1.43 | 0.33 | −4.3 | <0.001 |
| | | 25th | 3.69 | 0.41 | −9.02 | <0.001 |
| | | 50th | −23.85 | 0.76 | 31.28 | <0.001 |
| | | 75th | −1 | 0.31 | 3.22 | <0.001 |
| | | 90th | −0.33 | 0.15 | 2.13 | <0.001 |
| | High | 10th | 10.02 | 0.4 | 24.8 | <0.001 |
| | | 25th | 12.64 | 0.61 | 20.7 | <0.001 |
| | | 50th | 3.24 | 0.85 | 3.8 | <0.001 |
| | | 75th | 0.63 | 0.2 | 3.09 | <0.001 |
| | | 90th | 0 | 0.14 | 0 | **1** |
| | Low | 10th | 1.01 | 0.45 | 2.25 | **0.02** |
| | | 25th | 12.13 | 1.96 | 6.18 | <0.001 |
| | | 50th | 9.24 | 0.72 | 12.8 | <0.001 |
| | | 75th | 1.77 | 0.18 | 9.62 | <0.001 |
| | | 90th | 0.92 | 0.12 | 7.43 | <0.001 |
| Evening bat | Combined | 10th | 2.75 | 1.52 | −1.81 | **0.07** |
| | | 25th | 1.64 | 1.07 | −1.53 | **0.13** |
| | | 50th | 1.41 | 0.99 | −1.42 | **0.16** |
| | | 75th | 5.29 | 2.16 | −2.45 | 0.01 |
| | | 90th | 2.01 | 1.49 | −1.35 | **0.18** |
| | High | 10th | 2.43 | 2.11 | 1.15 | **0.25** |
| | | 25th | 3.87 | 1.05 | 3.69 | <0.001 |
| | | 50th | 10.29 | 1.15 | 8.93 | <0.001 |
| | | 75th | 14.27 | 1.29 | 11.09 | <0.001 |
| | | 90th | 6.53 | 1.42 | 4.6 | <0.001 |
| | Low | 10th | −1.02 | 1.04 | −0.98 | **0.33** |

(Continued)

| Table 1 (continued) | | | | | | |
|---|---|---|---|---|---|---|
| Species | Treatment | Percentile | Beta | SE | t-value | p |
| | | 25th | −7.65 | 2.17 | −3.53 | <0.001 |
| | | 50th | 1.72 | 1.55 | 1.11 | **0.27** |
| | | 75th | 7.51 | 1.72 | 4.36 | <0.001 |
| | | 90th | 1.41 | 1.57 | 0.9 | **0.37** |
| Brazilian free-tailed bat | Combined | 10th | −1.1 | 0.61 | −1.8 | **0.07** |
| | | 25th | −0.97 | 0.68 | −1.44 | **0.15** |
| | | 50th | −4.15 | 1.55 | −2.68 | 0.01 |
| | | 75th | 0.23 | 4.24 | 0.05 | **0.96** |
| | | 90th | 3.19 | 3.12 | 1.02 | **0.31** |
| | High | 10th | −5.13 | 0.62 | −8.23 | <0.001 |
| | | 25th | −1.99 | 1.27 | −1.57 | **0.12** |
| | | 50th | 0.3 | 2.35 | 0.13 | **0.9** |
| | | 75th | 4.6 | 6.45 | 0.71 | **0.48** |
| | | 90th | 9.63 | 6.36 | 1.52 | **0.13** |
| | Low | 10th | −3.52 | 0.51 | −6.95 | <0.001 |
| | | 25th | −1.2 | 0.75 | −1.6 | **0.11** |
| | | 50th | 17.08 | 1.98 | 8.6 | <0.001 |
| | | 75th | 18.19 | 2.86 | 6.35 | <0.001 |
| | | 90th | 13.16 | 2.65 | 4.96 | <0.001 |
| Tricolored bat | Combined | 10th | −41.58 | 0.79 | 52.35 | <0.001 |
| | | 25th | −43.18 | 0.75 | 57.76 | <0.001 |
| | | 50th | −34.97 | 2.89 | 12.11 | <0.001 |
| | | 75th | −8.51 | 1.59 | 5.35 | <0.001 |
| | | 90th | −0.37 | 0.82 | 0.45 | **0.65** |
| | High | 10th | −39.11 | 1.28 | −30.57 | <0.001 |
| | | 25th | −32.36 | 3.49 | −9.27 | <0.001 |
| | | 50th | −22.07 | 2.42 | −9.13 | <0.001 |
| | | 75th | −5.71 | 2.11 | −2.71 | <0.001 |
| | | 90th | −1.14 | 0.84 | −1.36 | **0.17** |
| | Low | 10th | −43.81 | 0.72 | −60.43 | <0.001 |
| | | 25th | −46.12 | 0.62 | −74.49 | <0.001 |
| | | 50th | −46.01 | 0.66 | −69.82 | <0.001 |
| | | 75th | −41.42 | 1.14 | −36.35 | <0.001 |
| | | 90th | −23.33 | 1.69 | −13.78 | <0.001 |

**Note:**

Beta values, standard errors (SE), t-values, and p-values from quantile regression analyses comparing flight distance during three treatment emissions from the NRG Systems ultrasonic deterrent (Low: 20, 26, and 32 kHz; High: 38, 44, and 50 kHz; Combined: 20, 26, 32, 38, 44, and 50 kHz) to a control period of no emissions for one species group, red bats (*Lasiurus borealis* and *Lasiurus blossevillii*), and four bat species cave myotis (*Myotis velifer*), Brazilian free-tailed bats (*Tadarida brasiliensis*), evening bats (*Nycteceius humeralis*), and tricolored bats (*Perimyotis subflavus*). The estimated effects are averages across individual bats for the different quantiles of control minus treatment differences in distance. Trials were conducted in a flight cage from 2020 to 2021 in San Marcos, Texas, USA. We assessed differences in flight distances using quantile regression and focused on the 10th, 25th, 50th, 75th, and 90th percentiles. Bold values indicate no significant difference in flight distance between treatment and control. $P < 0.016$ are considered to have $\alpha = 0.05$ with a Bonferroni correction for the three treatment comparisons.

**Table 2 Mean difference in distances flown by bats between control and ultrasonic emissions.**

| Species | Treatment | 10th | 25th | 50th | 75th | 90th |
|---|---|---|---|---|---|---|
| Red bats | Combined | 13.21 ± 15.60 | 19.57 ± 22.2 | 16.07 ± 21.27 | 7.81 ± 16.6 | 3.57 ± 12.29 |
| | High | 14.59 ± 15.77 | 18.83 ± 16.58 | 15.11 ± 15.47 | 7.32 ± 13.02 | 3.72 ± 8.79 |
| | Low | 9.01 ± 15.75 | 14.82 ± 18.63 | 13.29 ± 18.22 | 7.12 ± 15.43 | 3.14 ± 9.57 |
| Cave myotis | Combined | 11.71 ± 12.99 | 14.43 ± 15.45 | 9.75 ± 18.21 | 3.08 ± 8.64 | 0.86 ± 2.54 |
| | High | 7.28 ± 10.12 | 12.12 ± 15.27 | 8.71 ± 19.90 | 2.74 ± 8.34 | **0.89 ± 2.44** |
| | Low | **13.22 ± 15.14** | 16.66 ± 17.11 | 11.77 ± 16.10 | 2.70 ± 9.42 | 0.64 ± 3.58 |
| Evening bat | Combined | **2.06 ± 11.53** | **3.24 ± 16.89** | **4.23 ± 20.49** | 1.24 ± 17.65 | **−1.48 ± 13.19** |
| | High | **12.28 ± 15.96** | 13.59 ± 18.13 | 9.62 ± 18.00 | 4.97 ± 16.35 | 0.46 ± 11.68 |
| | Low | **4.19 ± 14.66** | 6.30 ± 19.24 | **3.36 ± 21.37** | 0.62 ± 18.73 | **−2.63 ± 15.34** |
| Brazilian free-tailed bat | Combined | **6.82 ± 14.28** | **9.05 ± 16.84** | 7.42 ± 14.87 | **3.89 ± 10.19** | **1.90 ± 4.57** |
| | High | 6.96 ± 14.16 | **7.78 ± 17.16** | **5.88 ± 15.97** | **2.00 ± 12.40** | **−0.23 ± 9.21** |
| | Low | 6.04 ± 16.01 | **5.97 ± 18.76** | 5.32 ± 17.45 | 1.70 ± 13.80 | −0.16 ± 10.39 |
| Tricolored bats | Combined | 10.34 ± 27.58 | 13.69 ± 30.08 | 12.8 ± 24.84 | 8.51 ± 17.88 | **1.61 ± 10.43** |
| | High | 5.94 ± 17.47 | 7.83 ± 19.89 | 10.92 ± 19.80 | 6.54 ± 15.72 | −0.34 ± 11.48 |
| | Low | 5.72 ± 20.12 | 11.40 ± 26.59 | 11.53 ± 25.63 | 7.63 ± 21.26 | **0.97 ± 10.21** |

**Note:**
Mean (±SD) difference in distances flown during three treatment emissions from the NRG Systems ultrasonic deterrent (UD) (Low: 20, 26, and 32 kHz; High: 38, 44, and 50 kHz; Combined: 20, 26, 32, 38, 44, and 50 kHz) compared to a control period of no emissions for one species group, red bats (*Lasiurus borealis* and *Lasiurus blossevillii*), and four bat species cave myotis (*Myotis velifer*), Brazilian free-tailed bats (*Tadarida brasiliensis*), evening bats (*Nycteceius humeralis*), and tricolored bats (*Perimyotis subflavus*). The estimated effects are averages across individual bats for the different quantiles of control minus treatment differences in distance. Trials were conducted in a flight cage from 2020–2021 in San Marcos, Texas, USA. Bold values indicate no significant differences between treatment and control during the quantile regression analysis that focused on the 10th, 25th, 50th, 75th, and 90th percentiles.

The differences in effectiveness of UDs among bat species could potentially be attributed to variation in echolocation behavior. Echolocation frequency and use varies across species (*Schnitzler, Moss & Denzinger, 2003*; *Jones & Holderied, 2007*); thus, species-specific responses should not be surprising. A study in south Texas documented that the NRG UD reduced mortalities of hoary and Brazilian free-tailed bats, both of which have lower echolocation frequencies. In contrast, mortalities of species with higher echolocation frequencies, such as the northern yellow bat (*L. intermedius*) were not significantly reduced (*Weaver et al., 2020*). Similar studies using other deterrent technologies have also reported varying results among species (*Arnett et al., 2013*; *Romano et al., 2019*). The effectiveness of UDs may result from the more rapid attenuation of high-frequency sound (*Griffin, 1971*). Deterrent signals that include lower-frequency ultrasound travel farther from the source and may be detected by bats at greater distances. However, our results did not indicate a trend in effectiveness based on echolocation frequency, as Brazilian free-tailed bats were the lowest-frequency bats tested and cave myotis were the highest-frequency echolocators, and both of these species had similar flight responses during high and low treatment emissions.

Results provided some evidence that bat species' responses may differ between the sexes, as we found to be the case for red bats and cave myotis; however, additional research focused on differences between the sexes is needed. An accurate understanding of the potential differences in wind turbine mortalities and UD effectiveness between sexes also is needed to modify and inform further deployment of UDs. Changes to the population sex

| Species | Treatment | Percentile | Sex | Season | Sex:Season |
|---|---|---|---|---|---|
| **Table 3 Results from analysis of variance assessments.** | | | | | |
| Red bats ($n$ = 46; sex df = 1; season df = 1; sex:season df = 1; residual df = 321) | | | | | |
| | Combined | 10th | 0.061 | <0.001 | 0.436 |
| | | 25th | 0.064 | <0.001 | 0.388 |
| | | 50th | 0.02 | <0.001 | 0.348 |
| | | 75th | 0.026 | <0.001 | 0.075 |
| | | 90th | 0.024 | <0.001 | 0.48 |
| | Low | 10th | 0.395 | <0.001 | 0.002 |
| | | 25th | 0.534 | <0.001 | 0.009 |
| | | 50th | 0.448 | <0.001 | 0.003 |
| | | 75th | 0.492 | <0.001 | <0.001 |
| | | 90th | 0.597 | <0.001 | 0.001 |
| | High | 10th | 0.423 | <0.001 | 0.082 |
| | | 25th | 0.512 | <0.001 | 0.136 |
| | | 50th | 0.418 | <0.001 | 0.139 |
| | | 75th | 0.556 | <0.001 | 0.022 |
| | | 90th | 0.593 | <0.001 | 0.261 |
| Cave myotis ($n$ = 57; sex df = 1; season df = 1; residual df = 396) | | | | | |
| | Combined | 10th | 0.668 | 0.792 | |
| | | 25th | 0.856 | 0.861 | |
| | | 50th | 0.894 | 0.991 | |
| | | 75th | 0.887 | 0.867 | |
| | | 90th | 0.56 | 0.983 | |
| | Low | 10th | 0.086 | 0.553 | |
| | | 25th | 0.042 | 0.762 | |
| | | 50th | 0.068 | 0.508 | |
| | | 75th | 0.019 | 0.315 | |
| | | **90th** | **0.015** | **0.413** | |
| | High | **10th** | **11th** | **12th** | |
| | | 25th | 0.737 | 0.857 | |
| | | 50th | 0.606 | 0.805 | |
| | | 75th | 0.824 | 0.669 | |
| | | **90th** | **0.997** | **0.88** | |
| Evening bat ($n$ = ; sex df = 1; season df = 1; residual df = 368) | | | | | |
| | Combined | **10th** | **0.754** | **0.42** | |
| | | **25th** | **0.917** | **0.479** | |
| | | **50th** | **0.82** | **0.493** | |
| | | 75th | 0.903 | 0.344 | |
| | | **90th** | **0.806** | **0.459** | |
| | Low | **10th** | **0.396** | **0.347** | |
| | | 25th | 0.298 | 0.758 | |
| | | **50th** | **0.306** | **0.521** | |
| | | 75th | 0.663 | 0.688 | |

| Species | Treatment | Percentile | Sex | Season | Sex:Season |
|---|---|---|---|---|---|
| | | **Table 3 (continued)** | | | |
| | | **90th** | **0.827** | **0.456** | |
| | High | **10th** | **0.177** | **0.108** | |
| | | 25th | 0.199 | 0.095 | |
| | | 50th | 0.13 | 0.162 | |
| | | 75th | 0.223 | 0.074 | |
| | | 90th | 0.309 | 0.103 | |
| Brazilian free-tailed bats (*n* = 73; sex df = 1; season df = 1; sex:season df = 499) | | | | | |
| | Combined | **10th** | **0.243** | **0.659** | **<0.001** |
| | | **25th** | **0.315** | **0.208** | **<0.001** |
| | | 50th | 0.326 | 0.259 | <0.001 |
| | | **75th** | **0.882** | **0.277** | **<0.001** |
| | | **90th** | **0.659** | **0.333** | **<0.001** |
| | Low | 10th | 0.036 | 0.452 | 0.078 |
| | | **25th** | **0.039** | **0.1** | **0.025** |
| | | 50th | 0.056 | 0.095 | 0.032 |
| | | 75th | 0.112 | 0.162 | 0.145 |
| | | 90th | 0.128 | 0.208 | 0.22 |
| | High | 10th | 0.065 | 0.641 | 0.023 |
| | | **25th** | **0.073** | **0.696** | **0.007** |
| | | **50th** | **0.074** | **0.464** | **0.004** |
| | | **75th** | **0.192** | **0.4** | **0.022** |
| | | **90th** | **0.365** | **0.699** | **0.084** |
| Tricolored bats (*n* = 17; sex df = 1; season df = 1; residuals df = 116) | | | | | |
| | Combined | 10th | 0.839 | 0.119 | |
| | | 25th | 0.709 | 0.041 | |
| | | 50th | 0.698 | 0.059 | |
| | | 75th | 0.931 | 0.057 | |
| | | **90th** | **0.724** | **0.066** | |
| | Low | 10th | 0.151 | 0.009 | |
| | | 25th | 0.163 | 0.008 | |
| | | 50th | 0.152 | 0.009 | |
| | | 75th | 0.187 | 0.001 | |
| | | 90th | 0.312 | <0.001 | |
| | High | 10th | 0.638 | 0.004 | |
| | | 25th | 0.435 | 0.003 | |
| | | 50th | 0.41 | 0.003 | |
| | | 75th | 0.553 | <0.001 | |
| | | **90th** | **0.953** | **<0.001** | |

**Note:**
Results from the analysis of variance assessments for pairwise comparisons of flight distance between the NRG System ultrasonic deterrent emissions and the control period of no emissions by sex, season, and sex within season for each bat species group (red bats (*Lasiurus borealis* and *Lasiurus blossevillii*)), or bat species (cave myotis (*Myotis velifer*), Brazilian free-tailed bats (*Tadarida brasiliensis*), evening bats (*Nycteceius humeralis*), and tricolored bats (*Perimyotis subflavus*)). Bold values indicate no significant differences between treatment and control during the quantile regression analysis that focused on the 10th, 25th, 50th, 75th, and 90th percentiles. "df" = degrees of freedom. $P < 0.016$ are considered to have $\alpha = 0.05$ with a Bonferroni correction for the three treatment comparisons.

**Table 4 Pairwise comparisons between sex, species, and sex within species.**

| Species | Treatment | Percentile | Pairwise results |
|---|---|---|---|
| Red bats | Combined | 10th | Spring > fall 9.339 m (<0.001) |
| | | 25th | Spring > fall 9.636 m (<0.001) |
| | | 50th | Female > male 4.576 m; spring > fall 8.810 m (<0.001) |
| | | 75th | Female > male 4.237 m (0.026); spring > fall 8.475 m (<0.001) |
| | | 90th | Female > male 4.169 m (0.024); spring > fall 7.585 m (<0.001) |
| | Low | 10th | Female spring > female fall 6.948 m (0.011); male spring > female fall 17.427 m (<0.001); female spring > male fall 8.709 m (<0.001); male spring > male fall 19.188 m (0.001); male spring > female spring 10.479 m (0.009) |
| | | 25th | Female spring > female fall 7.047 m (0.025); male spring > female spring 17.095 m (<0.001); female spring > male fall 8.342 m (0.003); male spring > male fall 18.390 m (<0.001); male spring > female spring 10.047 m (0.033) |
| | | 50th | Male spring > female fall 16.319 m (<0.001); female spring > male fall 7.910 m (0.003); male spring > male fall 18.413 m (<0.001); male spring > female spring 10.503 m (0.017) |
| | | 75th | Male spring > female fall 17.297 m (<0.001); female spring > male fall 7.190 m (0.004); male spring > male fall 20.284 m (<0.001); male spring > female spring 13.093 m (<0.001) |
| | | 90th | Male spring > female fall 15.244 m (<0.001); female spring > male fall 6.641 m (0.006); male spring > male fall 16.955 m (<0.001); male spring > female spring 10.313 m (0.006) |
| Red bats | High | 10th | Spring > fall 7.890 m (<0.001) |
| | | 25th | Spring > fall 8.481 m (<0.001) |
| | | 50th | Spring > fall 7.682 m (<0.001) |
| | | 75th | Male spring > female fall 12.295 m (<0.001); female spring > male fall 6.110 m (0.023); male spring > male fall 13.664 m (<0.001) |
| | | 90th | Spring > fall 6.013 m (<0.001) |
| Cave myotis | Low | 75th | Males > females 3.174 m |
| | | **90th** | **Males > females 3.176 m** |
| Brazilian free-tailed bat | Combined | **10th** | **Female fall > female spring 5.014 m (0.028); male spring > female spring 9.609 m (<0.001)** |
| | | **25th** | **Female fall > female spring 6.746 m (0.003); male spring > female spring 10.102 m (<0.001)** |
| | | 50th | Female fall > female spring 6.477 m (0.003); male spring > female spring 10.045 m (<.001) |
| | | **75th** | **Female fall > female spring 5.409 m (0.011); male spring > female spring 7.314 m (0.013)** |
| | | **90th** | **Female fall > female spring 4.745 m (0.022); male spring > female spring 5.872 m (0.05)** |
| | Low | 10th | Males > females 2.862 m (0.036) |
| | | **25th** | **Female fall > female spring 5.794 m (0.035); male fall > female spring 6.224 m (0.011); male spring > female spring 7.94 m (0.029)** |
| | | 50th | Female fall > female spring 5.52 m (0.039); male fall > female spring 5.810 m (0.015); male spring > female spring 7.243 m (0.045) |
| Brazilian free-tailed bat | High | 10th | Male spring > female spring 7.601 m (0.018) |
| | | **25th** | **Male spring > female spring 8.688 m (0.009)** |
| | | **50th** | **Female fall > female spring 4.715 m (0.079); male spring > female spring 8.594 m (0.006)** |
| | | **75th** | **No pairwise differences** |
| Tricolored bat | Combined | 25th | Fall > spring 9.262 m (0.046) |
| | Low | 10th | Fall > spring 11.098 m (0.010) |
| | | 25th | Fall> spring 11.591 m (0.010) |

| Species | Treatment | Percentile | Pairwise results |
|---|---|---|---|
| | | 50th | Fall > spring 10.987 m (0.011) |
| | | 75th | Fall > spring 12.430 m (0.002) |
| | | 90th | Fall > spring 12.422 m (0.001) |
| | High | 10th | Fall > spring 10.929 m (0.004) |
| | | 25th | Fall > spring 12.063 m (0.004) |
| | | 50th | Fall > spring 11.426 m (0.004) |
| | | 75th | Fall > spring 11.387 m (0.001) |
| | | **90th** | **Fall > spring 10.97 (<0.001)** |

**Note:**
Pairwise comparisons from analysis of variance of flight distance between the NRG System ultrasonic deterrent emissions and the control period of no emissions by sex, season, and sex within season for each bat species group (red bats (*Lasiurus borealis* and *Lasiurus blossevillii*)), or bat species (cave myotis (*Myotis velifer*), Brazilian free-tailed bats (*Tadarida brasiliensis*), evening bats (*Nycteceius humeralis*), and tricolored bats (*Perimyotis subflavus*)). Bold values indicate no significant differences between treatment and control during the quantile regression analysis that focused on the 10th, 25th, 50th, 75th, and 90th percentiles.

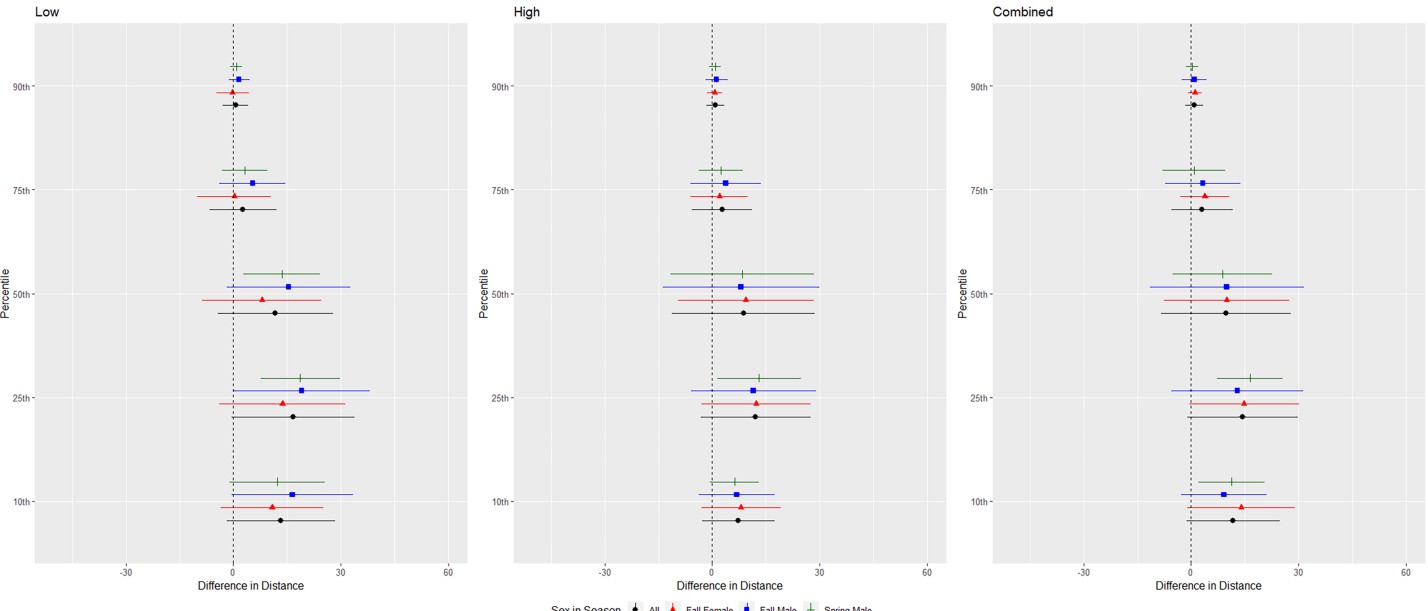

**Figure 6 Differences in flight distances that cave myotis (*Myotis velifer*) flew during three ultrasonic deterrent emissions *vs.* a control period.** The differences in distance (m) that cave myotis (*Myotis velifer*) flew from the ultrasonic deterrent (UD) during each emission treatment (Combined, Low, High) *vs.* the control period with the UD powered off by sex, season, and sex within season.

ratio can greatly influence population growth, size, and risk of extinction (*e.g.*, *Donald, 2007*; *Lehikoinen et al., 2008*; *Wedekind, 2012*; *Ramula et al., 2018*), as having too few females can limit population growth. The importance of females to population growth and stability is particularly true for bats because many species have polygamous mating systems, and females only have one litter per year and typically fewer than two pups per litter (*Barclay et al., 2003*; *Ammerman et al., 2019*). Therefore, activities that reduce the relative abundance of females are likely to lead to more dramatic population declines (*Wedekind, 2012*). Thus, minimization strategies that target females during periods of high

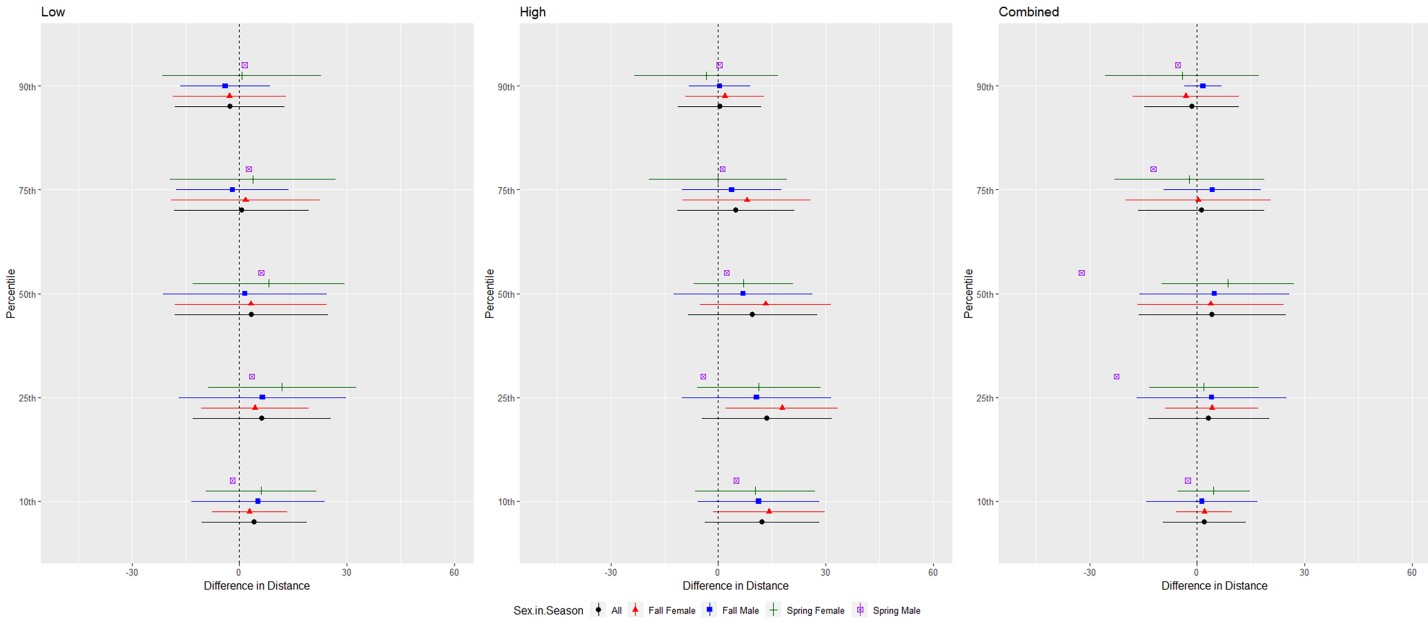

**Figure 7 Differences in flight distances that evening bats (*Nycteceius humeralis*) flew during three ultrasonic deterrent emissions *vs.* a control period.** The differences in distance (m) that evening bats (*Nycteceius humeralis*) flew from the ultrasonic deterrent (UD) during each emission treatment (Combined, Low, High) *vs.* the control period with the UD powered off by sex, season, and sex within season.

risk may be more cost-effective and may provide similar population-level results as those targeting both sexes.

Results also indicate that UD effectiveness could potentially differ between spring and fall for some bat species, but, again, additional research with more seasons is warranted. For example, our exploratory analysis suggested that tricolored bats flew farther from the UD compared to the control during fall, whereas we observed greater flight distances for red bats during spring. Previous studies often focused UD testing in the late summer through fall seasons (*Szewczak & Arnett, 2007*; *Johnson et al., 2012*; *Arnett et al., 2013*; *Romano et al., 2019*; *O'Neil, 2020*; *Weaver et al., 2020*) because this is when bat mortalities peak at wind energy facilities in North America (*Arnett et al., 2008*; *Zimmerling & Francis, 2016*; *American Wind Wildlife Institute, 2021*). A recent study by *Goldenberg et al. (2021)* used thermal video data to show that bats spend more time flying near wind turbines and exhibit riskier behavior in late summer and fall. It is unclear, however, why bats spend less time near wind turbines during spring and early summer (*Drake, Schumacher & Sponsler, 2012*, *Kerns & Kerlinger, 2004*). Increasing evidence suggests that bats are attracted to wind turbines (*e.g.*, *Foo et al., 2017*; *Richardson et al., 2021*; *Guest et al., 2022*), which could, in part, explain the lack of predictive relationship between indicators of risk preconstruction and estimates of bat mortality postconstruction (*e.g.*, *Lintott et al., 2016*; *Solick et al., 2020*). A variety of explanatory hypotheses for bat attraction to wind turbines have been proposed (*Cryan & Barclay, 2009*; *Guest et al., 2022*), none of which are mutually exclusive and all of which likely vary with factors such as season, food availability, and reproductive condition. In this study, however, we can rule out any influence of attraction, as there was not a wind

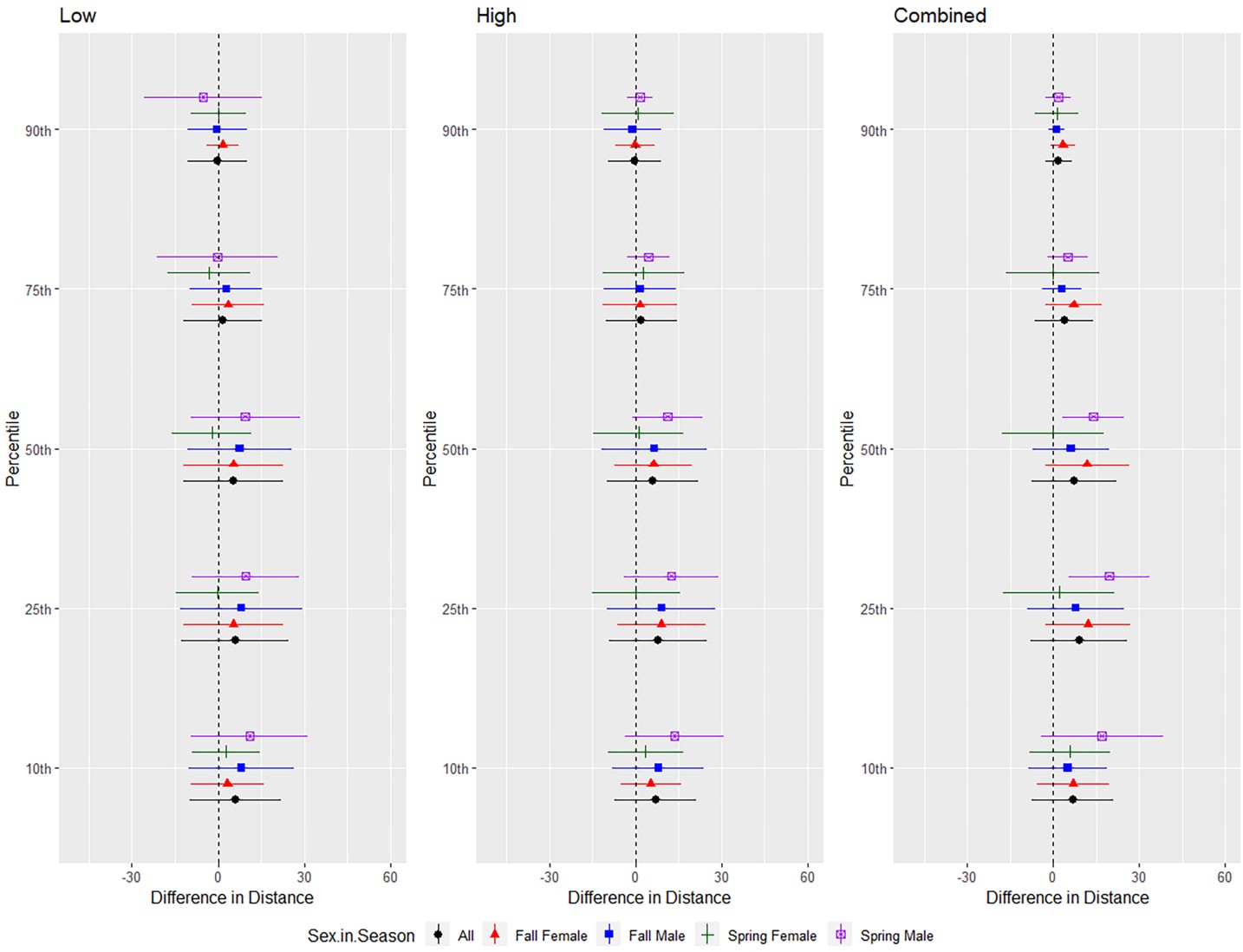

**Figure 8 Differences in flight distances that Brazilian free-tailed bats (*Tadarida brasiliensis*) flew during three ultrasonic deterrent emissions *vs.* a control period.** The differences in distance (m) that Brazilian free-tailed bats (*Tadarida brasiliensis*) flew from the ultrasonic deterrent (UD) during each emission treatment (Combined, Low, High) *vs.* the control period with the UD powered off by sex, season, and sex within season.

turbine or other large structure present in the immediate vicinity of the flight cage, and the end of the flight cage from which the UD was deployed was randomly selected each night.

More evidence is needed to understand differences in effectiveness between seasons and a potential biologically meaningful result in reducing mortalities. If a seasonal component to UD effectiveness exists, particularly with female bats, or if there is a window of time in which more females than males of a given species are being killed, then impact minimization strategies focused on that period would have a greater positive effect on population stability than strategies focused on time periods with greater risk to males. Much progress has been made in describing patterns of bat mortality related to wind turbines (*Arnett & Baerwald, 2013*; *Guest et al., 2022*). For example, a once widely held

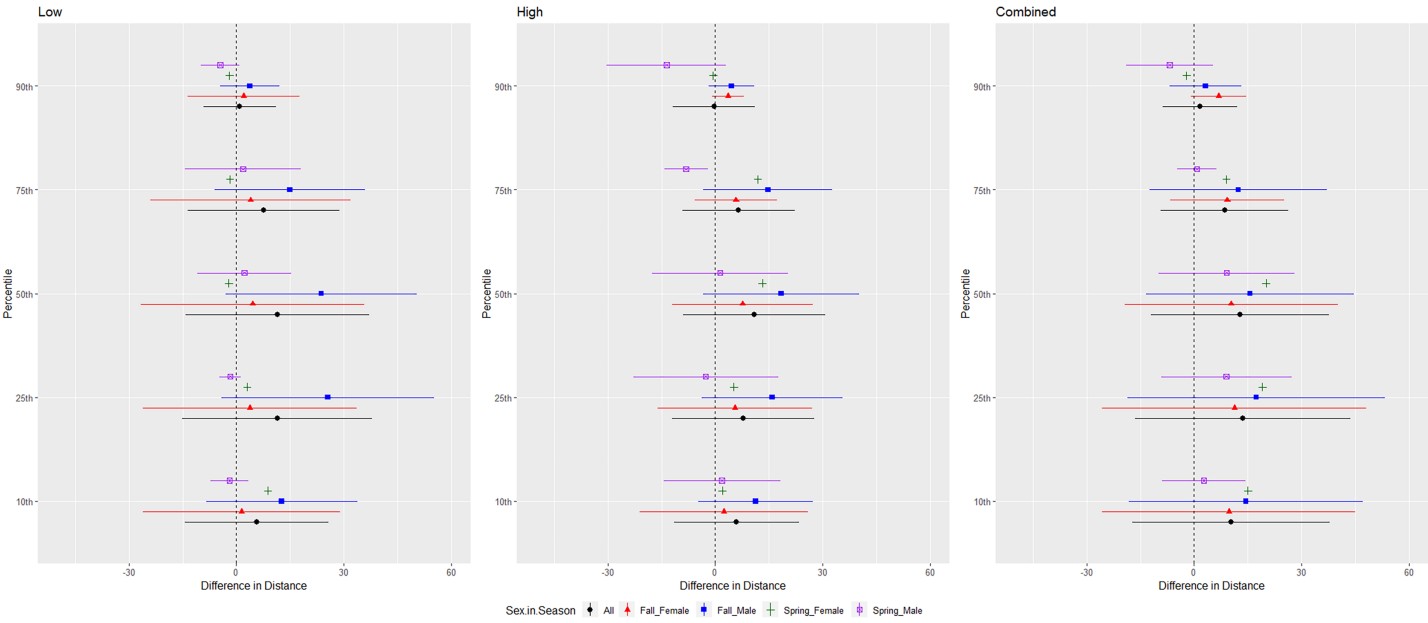

**Figure 9 Differences in flight distances that tricolored bats (*Perimyotis subflavus*) flew during three ultrasonic deterrent emissions *vs.* a control period.** The differences in distance (m) that tricolored bats (*Perimyotis subflavus*) flew from the ultrasonic deterrent (UD) during each emission treatment (Combined, Low, High) *vs.* the control period with the UD powered off by sex, season, and sex within season.

assumption within the wind-wildlife community was that relatively more male than female bats are killed at wind energy facilities in North America. Empirical support for this assumption came from morphological sex identification of bat carcasses collected in the field (*e.g.*, *Johnson et al., 2003*; *Fiedler, 2004*; *Arnett et al., 2008*). However, more recent genetic-based sex identifications indicate that morphology-based sex identifications of carcasses are inaccurate and often significantly overestimate the relative abundance of males (*Korstian et al., 2013*; *Nelson et al., 2018*; *Chipps et al., 2020*) because relatively more females are either misidentified or classified as unknown sex (*Korstian et al., 2013*; *Nelson et al., 2018*).

We identified several strengths and limitations in our study. First, this is the first study to examine species-specific differences of bats to various UD treatments in a semi-controlled environment (*i.e.*, an outdoor flight cage). With this facility, we could visualize the responses of individual bats of known species to different acoustic treatments using statistically robust methods. With the exception of the mesh netting, the flight cage environment was as similar as possible to what the local-caught bats were experiencing just prior to the experimental trials. However, because we were interested in examining how bats would respond to UD emissions in the real world, we did not control for the effects of weather, which could affect sound attenuation and/or bat behavior. Also, the observer location changed depending on the side with the active UD; however, this did not have observable effects on bat behavior. The main limitation was the length of the flight cage. Although it is longer than the blade length of most land-based wind turbines currently deployed, it did restrict flight to within 60.2 m of the UD and cannot account for increases

in blade length. Thus, the UDs may have been more effective than our results suggest, and if the flight cage were longer, it is possible that further differences among treatments could be detected. Lastly, the two-dimensional cameras limited the ability to estimate depth of the bat during flying; thus, we assumed bats were flying on the center line of the flight cage. Given that a bat could actually be within 3 m of the center line, the average uncertainty across the X positions of the bat is 43 cm from the deterrent.

Other published studies on UDs have primarily focused on using bat carcasses to estimate and compare mortalities among control and treatment conditions (*e.g.*, *Arnett et al., 2013*; *Romano et al., 2019*; *Weaver et al., 2020*), which does not allow researchers to incorporate behavioral observations of individual bats in the presence of UDs. A few studies have also tested the responses of free-flying bats to UDs by using thermal cameras over ponds or in riparian areas or at wind turbine towers, where species and/or sex could not be determined (*Johnson et al., 2012*; *Lindsey, 2016*; *Gilmour et al., 2020*; *Gilmour et al., 2021*). In these cases, the researchers could assess how the bat community responded to the ultrasonic broadcast by using thermal or night vision cameras, but they could not make inferences to sex or individual bat species. Cameras recording flight behavior of bats at wind turbines cannot yet provide information for species identification (*e.g.*, *Horn, Arnett & Kunz, 2008*).

For future testing, we recommend programming UDs to focus only on relatively low-frequency ultrasound (*e.g.*, <40 kHz). This range covers other species that are vulnerable to wind energy development, such as hoary bats and silver-haired bats. We also suggest exploring the use of frequency sweeps or different sound patterns, such as randomized pulsed signals. Complex signals may further disorient bats who might adapt to constant stimuli. For future experiments using a flight cage, we suggest extending the length of the flight cage from 60 m to at least 100 m to account for longer turbine blades. We also recommend randomly assigning the UD that emits the deterrent signal among treatments.

## CONCLUSIONS

This study demonstrated that certain bat species respond to different ultrasonic treatments, suggesting UDs could be a viable method for reducing bat fatalities at wind energy facilities, but variability among species existed. We observed similar results regardless of treatment for low-frequency and high-frequency bats. However, the low or combined treatments were most effective for the combined red bat species group, Brazilian free-tailed bats, cave myotis, and tri-colored bats, but not for evening bats. Furthermore, lower-frequency sounds attenuate less quickly and can cover a larger volume of airspace around a wind turbine. Thus, we suggest assessing the effectiveness of low frequency emissions *in situ*.

## ACKNOWLEDGEMENTS

The success of this project depended on assistance from the land manager of the Freeman Center at Texas State University, C. Thomas, countless technicians including M. Moreno, R. Tyler, K. Smith, K. Dyer, A. Commiskey as well as countless volunteers, particularly to

assist in building the flight cage. We thank B. Cade for significant assistance with the statistical methods. We would like to thank the Associate Editor for assistance in manuscript preparation. This work was authored in part by the National Renewable Energy Laboratory, operated by Alliance for Sustainable Energy, LLC, for the U.S. Department of Energy (DOE) under Contract No. DE-AC36-08GO28308. Funding provided by U.S. Department of Energy Office of Energy Efficiency and Renewable Energy Wind Energy Technologies Office and TCU-NEER Research Partnership. The views expressed in the article do not necessarily represent the views of the DOE or the U.S. Government. The U.S. Government retains and the publisher, by accepting the article for publication, acknowledges that the U.S. Government retains a nonexclusive, paid-up, irrevocable, worldwide license to publish or reproduce the published form of this work, or allow others to do so, for U.S. Government purposes.

### Funding

Funding was provided by U.S. Department of Energy Office of Energy Efficiency and Renewable Energy Wind Energy Technologies Office. The funders had no role in study design, data collection and analysis, decision to publish, or preparation of the manuscript.

### Grant Disclosures

The following grant information was disclosed by the authors:
U.S. Department of Energy Office of Energy Efficiency and Renewable Energy Wind Energy Technologies Office.

### Competing Interests

Sara Patricia Weaver & Emma Elizabeth Guest are employed by Bowman. Amanda Marie Hale is employed by Western EcoSystems Technology, Inc. Brogan Page Morton is employed by Wildlife Imaging System. Cris Daniel Hein is employed by National Renewable Energy Laboratory (operated by Alliance for Sustainable Energy, LLC.

### Author Contributions

- Sarah Rebecah Fritts conceived and designed the experiments, performed the experiments, analyzed the data, prepared figures and/or tables, authored or reviewed drafts of the article, and approved the final draft.
- Emma Elizabeth Guest performed the experiments, authored or reviewed drafts of the article, and approved the final draft.
- Sara P. Weaver conceived and designed the experiments, authored or reviewed drafts of the article, and approved the final draft.
- Amanda Marie Hale conceived and designed the experiments, authored or reviewed drafts of the article, and approved the final draft.
- Brogan Page Morton conceived and designed the experiments, authored or reviewed drafts of the article, and approved the final draft.

- Cris Daniel Hein conceived and designed the experiments, authored or reviewed drafts of the article, and approved the final draft.

## Animal Ethics

The following information was supplied relating to ethical approvals (*i.e.*, approving body and any reference numbers):

All work was approved by the Texas State University Institutional Animal Care and Use Committee (#664) and Texas Parks and Wildlife Department state permit (SPR-1217-243).

## Field Study Permissions

The following information was supplied relating to field study approvals (*i.e.*, approving body and any reference numbers):

Bat captures were approved by Texas State University Institutional Animal Care and Use Committee permit (#6224) and the Texas Parks and Wildlife Department state permit (SPR-1217-243).

## Data Availability

The raw data and R code are available in the Supplemental Files.

## Supplemental Information

Supplemental information for this article can be found online at http://dx.doi.org/10.7717/peerj.16718#supplemental-information.

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
