# Peer review of "Experimental trials of species-specific bat flight responses to an ultrasonic deterrent"

_PeerJ, doi:10.7717/peerj.16718_

## Round 0.1 · original submission · Major Revisions

There are a range of questions and suggested improvements. Please address the comments of the reviewers, especially make your work clearer and discuss comprehensively (by staying concise) the reviewers' criticisms. I hope you are willing to do that.

·

Basic reporting

Please see some comments in the line by line comments below on basic reporting. I have some criticisms about the way the methods are reported in general. This section could be made much more concise and headings would be helpful. The schematic is useful. There are some missing bits or ways it could be restructured so it is easy to pick out salient information.

Line by line comments. Please refer to the annotated review of the pdf for these comments and highlighted areas.

L100: Include Gilmour et al. 2021 https://doi.org/10.1242/jeb.242715 Jamming hypothesis is perhaps a bit simplified understanding now of how bats respond to UDs.

L112: Again, you fail to include the work done flight path tracking bats in response to UDs Gilmour et al 2021. Only one species tested, but a start in this area of work.

L128 This section could be made more concise and structured better. E.g. combine the fact that you are placing two UDs in the flight cage into the first paragraph and add details later. Otherwise you are sent to look at Figure 1 before you have the full info on experimental set up.

L128 Perhaps the addition of headers for the different methods sections would make this an easier and more organised read? If the journal allows?

L129 Over which dates, on how many nights? Multiple years. This is very important info. Temp and humidity controlled for or collected? Rainfall nights excluded? This section needs signicant improvement and clarification.

L140: State that the UDs were at either end of the flight cage, as in the figure.

L142 Why these specific frequencies? Were they chosen as you state in next paragraph? If so why step up by 6 kHz each time, some explanation would be helpful.
L143: Across frequencies?

Or just say "these frequencies, ranging from 20-50 kHz were selected...."

L144: Can you calculate a dB table for the F bands at a range of distances from the UDs? This may have a bearing on how a bat responds to the different F bands, as higher bands may attenuate more quickly, so the bat is not experiencing a uniform SPL for all bands.

L49: missing 'was'. Also this sentence maybe should be combined with the last, or include "for this species"?

L160 Two hours driving? A distance would be better in km?


L162 This info needs to come earlier and an indication of how many nights the experiment took place over. It all needs a bit of a restructure, so key info is covered earlier about experiments and details come later.

L63: human-induced?

L164 What was the reason for feeding the bats?

L165 Include the length of the different periods in the text as well as in the figure.

Also have you got any reasoning on the acclimation period length, or something to reference on this. It seems very short, but there may be a reason for this.

L171 Why? Include a reasoning for this.

L172 Good, but again why? Include a reasoning for this.

L178 What protocols?

L177 Can you be clearer here- why were bats injured? During capture, or in the experiment. This worries me. Were there any severely injured bats? If so, how many? This probably needs rewording, as sounds convoluted and unclear. Was it just you had permission to do this, had there been an injured bat, or you did have to do it?

L198 Did cameras have any distortion at edges? Was this accounted for?

L206 What was your reasoning for not using a GLMM or something similar? It seems that you may have an order affect of your C vs T periods, i.e. bats could become habituated, or some other order effect related to flight cage stress, or stress after handling.

Would it not make sense to use a random effects model?

Or this? https://cran.r-project.org/web/packages/lqmm/lqmm.pdf

I am really not convinced with your stats method. You will have to explain how you accounted for multiple comparisons and why you did not consider including treatment order as a random effect.

L208 Would you not need to include a correction for multiple comparisons here, e.g. Bonferroni or something similar?

L214 Did you compare just the first control period to the first treatment period, or to all treatment periods combined?

L220 Again, if you are doing multiple comparisons, you should really include a correction for this, as you will increase Type 1 error \(false positives\) the more tests you do.

You could also include all of these fixed effects and interactions in one big model, surely that would give you better power to detect effects and make more sense statistically?

L237 There has got to be a better way of presenting the data. These plots are hard to interpret and compare between species. A box plot with all species responses to each F band would be much better and maybe the sex and season \ info represented in a table or something? Or if you can get the sex and species in there too that would be great, if it wasn't too busy. I appreciate its difficult to tease apart, but this definitely needs work. The whole premise of the paper is comparing species-specific differences, but your analysis and presentation of the results doesn't seem to follow this.

L271 Discussion: A better way to structure this would be to include headings, maybe questions linked to key findings? E.g. Do bat species differ in their responses to UDs at distance? Or something similar...

L274 typo experimental

L273 be specific, shifting how? More precise language needed

L276 Maybe due to the analysis?

L296 This phrasing tends to suggest the deterrent has active part in "interacting" with individuals. This needs rephrasing. Also better to use individuals rather than species as the responding party. The attenuation of sound with increasing distance could be explained better. This is also dependent on atmospheric pressure and temperature. Were these considered for different nights of experiments?
I cannot see any mention of temp/humidity effects. At higher Fs humidity and temp can have a big effect on deterrent propagation over distance. This could have affected what the bats were experiencing. If the temp and humidity were fairly consistent across the experiment state so.

L298 It may not just be the echolocation peak F that determines how bats respond to different F bands from UDs. The call type should also be considered. Myotis bats tend to have a broader sweeping FM component compared to your “lower-frequency" bats. They therefore have in a sense, perhaps no where to go in changing their call structure in response to sound. Bats from species with a quasi CF part of their call, or other call structures may be able to modify their calls in response to a masking or "jamming" sound. See Gilmour et al 2021 again. This is crucial in understanding why bat species may respond differently to UD F bands. This needs more discussion. Reference to low or high F bats is too simplistic, as call structure and duration may also play a part in species-specific responses.
I am not familiar with the call structures of these bats, but it may be worth considering as a line of reasoning!

L301 Again a heading would really help here to break up the discussion and make it clearer what we are discussing.

L302 Any suggestions to why there may be a difference in responses of the sexes?
How do they differ? In what direction? This is too vague.

L329 So why could there be a difference here between season and species? Maybe some of these things going on here too. Link back to your results.

L338 This is an important point

L347 Interesting!

L350 Try not to spend too many words explaining the limitations. This bit can be cut down a bit as it reads more like a thesis or report than a paper. These bits could be incorporated into the methods somehow or cut down to a short concise paragraph. They are important points, but you have good reasoning to do it the way you did, so discussing limitations at the end reduces the impact of the study.

L362 And Gilmour et al. 2021

L387 This sentence needs to be reworded. "Shift activity" seems the wrong phrasing, since you have been using distance all the way through.

L385 This should just contain a punchy outline of the main conclusions. I would suggest removing the second paragraph and any limitation suggestions into the main discussion and keeping it concise and a confident round off of why your study is great, because it is! I feel you are not selling it enough.

L390 Again focusing on call type rather than species would be a better way to approach this study. There are standard ways of doing this in the literature. This would make it more applicable to international audiences looking at bats that have similar acoustic niches to those tested in your study.

L393 What's the point in the study then? You could reword this, saying something like "caution must be taken extrapolating these results to a wind turbine application" or "future studies should focus on how these \ results compare to a wind turbine in situ..." or something along those lines! This is a proof-of-concept for the overall application to WE scenarios, looking at specific behaviour. It's always going to be hard to extrapolate. But this phrase seems to suggest the experiment is pointless!
L398 Explain why... this might be better in the main discussion

Experimental design

While I have no problems with the experimental design in its basic form, the analysis performed needs some work, or more explanation. Please see my detailed comments in the line by line section.

Validity of the findings

Some work needs to be done to convince me of the findings. They also need to be presented more clearly and reanalysis may make this possible. If the results could be presented and discussed so they are applicable to a wider audience this would make the work much more accessible and the impact higher.

Additional comments

This is an important study in the field of deterrent research and seems to be a really good flight cage experiment, well thought out and designed, within the limits of possibilities. It is an important next step in understanding how ultrasonic devices can be used to reduce mortalities of bats at wind turbines. The manuscript however is lacking detail in places and a strong message concluded from the results is missing. I recommend an alternative analysis method and a better way of presenting and discussing results. These corrections would really improve the manuscript and significantly increase its impact in the field. It also needs some work making it applicable outside of the north American scenario. If bats were described based on their call characteristics or in a guild specific way perhaps, it would be easy to compare to European bats or those elsewhere where wind energy is expanding (India for example). I have strong reservations about the statistics and the way the analysis has been performed, especially with multiple comparisons and no correction applied (or reference to if it was applied). Major re-writing needs to take place and perhaps some reanalysis. But if this is done, I wholeheartedly would like to see this published. I can not recommend for publication without significant corrections.

Reviewer 2 ·

Basic reporting

This study investigated whether bats respond differently to various frequency band of ultrasound broadcast from an acoustic deterrent, and if any differential response by species. The manuscript was well written with only minor typos and suggested edits.

See suggested edits in attachment.

Experimental design

The study used a flight tent to enable a controlled environment and facilitate monitoring and data collection. The experimental design and results support the conclusions of differential species and frequency band responses.

Validity of the findings

Although the study design and results present a convincing case for differential effects of ultrasound broadcast by species and frequency band, the authors extend the conclusions to less supported, and speculative, conclusions regarding sex and seasonal differences. While the data do suggest a response for these variables, the authors should not present or speculate upon these responses as caused by the ultrasonic deterrent as they may result merely from other effects. The experimental design and analysis did not have controls to evaluate these effects independent of testing the response to the ultrasonic deterrent. Minimally, the analysis should evaluate the control flight behavior of each species by sex and season rather than just the differential control/experiment for each case. The authors should present the sex and season data, but not in a way of confusing causality with correlation in the absence of strict experimental design to test those effects. For example, statements such as, "Our results also indicate that UD effectiveness _CAN_ differ between spring and fall for some bat species." do not seem warranted from the experimental design and analysis presented.

In this regard, without establishing the causality of sex and seasonal changes the extent of the discussion devoted to this aspect seems unwarranted, as those effects may possibly occur at wind turbines with or without ultrasonic deterrents.

Unless the authors can revise the ms. to present a more convincing case for causality of the sex and season effects the authors should focus on the differential species and frequency band effects for which they have a convincing study design, analysis, and presentation, and tone down the narrative of sex and season effects to an intriguing potential for further investigation.

Additional comments

Figs. 4–8 present the data in a confusing and not easily interpretable way. Instead, the authors should cluster the different percentile results together for each class (i.e., sex and season), but use separate vertical lines with error bars for each class. This would make the differences across the percentiles and classes readily interpretable.

Annotated reviews are not available for download in order to protect the identity of reviewers who chose to remain anonymous.

·

Basic reporting

Bats are an important part of the ecosystem. They help in pollination as well as dispersing seeds. Therefore, safety of bats during wind farm operation should be as vital as renewable energy. This study was designed to understand the bat species-specific responses to ultrasonic deterrent (UD) in order to reduce their fatality from the wind farm operations. The paper is well written and clear. The study overcame the challenges associated with experiments with live biological subjects like bats to produce empirical data for the effectiveness of UD.
Therefore, the paper should be accepted with below mentioned revisions.

Experimental design

1) In line 105, the authors have mentioned that “species-specific variability in effectiveness are unknown” and “deterrent configuration” could be one of the factors. Subsequent literature review also shows the difference in effectiveness between NRG systems and GE Renewable Energy UDs. However, this factor has not been addressed or discussed for this study. It will be useful if the authors could relate the outcome of this study to GE Renewable Energy UD for wider application.

2) Was the observer position fixed or changed depending on the operating UD? Does this have any effect on the bat flight path?

3) How many bats were present in the cage during each trial?

4) Authors have not discussed the effect of wind farm noise on the effectiveness of the UD. It has been shown that bats can change their echolocation strategy based on the background noise [1]. Will the conclusion of this study change due to wind speed and noise generated by wind turbine?

5) Attenuation of ultrasound in air can be above 1 dB/m depending on the temperature and relative humidity of the air [2]. This can result in drastic drop (up to 30 dB or higher at 30 m distance) in UD signal. Especially in case of the prolonged experiment period the signal strength would have changed substantially due to change in weather and temperature. The study did not report the signal strength along the cage during the period. Will this affect the analysis?

Reference:
[1] Tressler, J. and Smotherman, M.S., 2009. Context-dependent effects of noise on echolocation pulse characteristics in free-tailed bats. Journal of Comparative Physiology A, 195, pp.923-934.

[2] Lawrence, B.D. and Simmons, J.A., 1982. Measurements of atmospheric attenuation at ultrasonic frequencies and the significance for echolocation by bats. The Journal of the Acoustical Society of America, 71(3), pp.585-590.

Validity of the findings

1) Authors should include schematic and pictures of the ultrasonic arrays, as part of the experimental set-up. Also, the UD model must be mentioned.

2) What is the uncertainty in the distance measurement due to the limitations (e.g., resolution of the image) of the camera? This may change the outcome of the experiment where the difference between the groups is small.

3) In Line 256, “Although the low treatment was significant at the 10th, 50th, 75th, and 90th percentiles, the difference in distances between treatment and control were low, and bats flew closer to the UD at the 90th percentile”.
There is no significant difference between 25th percentile and all other percentiles mentioned. Also, the difference between combined and low treatment is small, from Figure 7.
Authors must amend the observation in Line 256.

Additional comments

1) Line 143: Reference missing for “This frequency range was selected because it encompasses the echolocation range of most bat species known to occur in the United States and Canada.”

2) Line 164: ad libidum (or ad libitum, typo error?)

3) Figure numbers in Line 246, 252, 255, 260, 269 are wrong. They do not match with Figure labels. Need correction.

---

## Round 0.2 · Major Revisions

While the reviewers are in general happy with the responses, all comments need to be addressed. If there are new questions arising, please discuss these as well - just responding to the reviewer will not help the readers of the article if they have a the same or a similar question. Please stay concise and precise - a comment from the editorial point of view: you should read through and reduce redundancies with the main aim to make the article more succinct.

·

Basic reporting

I am happy with the basic reporting of the manuscript and thank the authors for the substantial changes made to structure and addition of information to make this manuscript as clear as possible.

General comments

A few minor suggestions and responses to the comments are added to the rebuttal letter, however I do not need to see again and will leave to the authors discretion. There are a number of typos due to track changes that we suggest the authors check before final publication. I may not have found them all.

Line 150: two words together, sure you would find that in the final edit, but just flagging for ease.
Line 167: two words together
Line 409: extra space and period

Experimental design

The methods and discussion sections are very well structured now and although I would prefer headings throughout, I can see that the work done has made it much clearer to read and find the necessary information.

Everything is clarified and I am satisfied that the authors have made a substantial effort based on my suggestions. I was particularly impressed by the explanation on the statistics, which I have reservations about. I am happy now that they have exhausted all possibilities and come up with the best methodology to gain the insights they were hoping for from the study. This is a brilliant study and I would be happy to see it published.

Validity of the findings

see previous comments

·

Basic reporting

Authors have responded to the most of the comments clearly. However, there are some comments from review1 that still require further explanation and discussion in the manuscript.

Experimental design

(A) Review1, comment(2): Was the observer position fixed or changed depending on the operating UD? Does this have any effect on the bat flight path?
Response1:
The observer position changed with the randomly-selected UD per night. This was mandatory as we had to be within ~20 m to control the UD from the switchboard.
Review2, comment:
The authors mentioned that the observer’s position changed with randomly selected UD every night. But does this have any noticeable effect in the flight of the bats? Authors must include this in the discussion, as this information is useful in designing experiments for future studies.


(B) Review1, comment(4): Authors have not discussed the effect of wind farm noise on the effectiveness of the UD. It has been shown that bats can change their echolocation strategy based on the background noise [1]. Will the conclusion of this study change due to wind speed and noise generated by wind turbine?
Response1:
The frequencies are different of wind turbine noise and the deterrent signals are different. Also, we can only conduct so many treatments and were not able to add 'turbine noise' to our study design. Thus, we cannot comment at this time if the other noises would change the conclusion. We did add that future studies should compare in situ testing of this UD configuration.
Review2, comment:
Authors must clearly mention this limitation of lack of in situ testing of UD to the discussion, as the effect of turbine noise and wind noise is not considered in this study.

(C) Review1, comment(5): Attenuation of ultrasound in air can be above 1 dB/m depending on the temperature and relative humidity of the air [2]. This can result in drastic drop (up to 30 dB or higher at 30 m distance) in UD signal. Especially in case of the prolonged experiment period the signal strength would have changed substantially due to change in weather and temperature. The study did not report the signal strength along the cage during the period. Will this affect the analysis?
Response1:
Thank you for the suggestion. This calculation has been completed before (see Weaver et al. 2020) and was acknowledged in the discussion (line 294)
Our focus was conducting trials during real-world conditions and to assess results from a variety of combinations of temperature and humidity, just as a bat would experience at a wind farm. Thus, we did not include the specific environmental variables into the analysis.
Review2, comment:
Authors have mentioned the calculations were completed in reference Weaver et al. 2020. As shown in Weaver et al. 2020 minimum 55 dB SPL is necessary to deter bats. Authors need to mention this criteria with respect to the current study and present the SPL for the current study.

Validity of the findings

(A) Review1, comment(1): Authors should include schematic and pictures of the ultrasonic arrays, as part of the experimental set-up. Also, the UD model must be mentioned.
Response1:
We included a schematic. The UD does not have a model number, just a manufacturer.
Review2, comment:
Schematic of the experimental set up is in Figure1. Authors should also include a schematic or picture of the ultrasonic array to show the design of the UD for future studies.

(B) Review1, comment(2): What is the uncertainty in the distance measurement due to the limitations (e.g., resolution of the image) of the camera? This may change the outcome of the experiment where the difference between the groups is small.
Response:
The pixel sample area at the center of the cage (28m from the camera) is 2.5cm x 2.5cm. At that pixel size each bat detection was comprised of many pixels and therefore we could resolve the centroid of the detection area to subpixel accuracy. Therefore the resolution contributes much less than 2.5cm error.
Reprojecting the 2D pixel location in the video to the distance from the end of the cage does contribute uncertainty to the measurement. Since we do not have a depth estimate of the bat from the 2D video, we assume that the bat was on the center line of the cage for the transform. Since the bat could be +/- 3 meter of the center of the cage, we can calculate the error associate with those two extremes. Given the geometry, the uncertainty is a function of the X position within the video frame. Using the extreme values the average uncertainty across the X positions in 43 cm of distance from the deterrent.
Review2, comment:
The above analysis must be included the discussion of the manuscript.

---

## Round 0.3 · accepted · Accept

Manuscript can be accepted and send for production.

·

Basic reporting

No comment

Experimental design

The authors have explained the difficulty in conducting such studies as too many environmental variables and to include the wind farm noise is not in the scope of this study. Therefore, no further explanations required. I am satisfied with the responses and I thank authors for the changes and explanations.

Validity of the findings

Authors have addressed the comment satisfactorily.

Additional comments

The authors have addressed all the comments and I am satisfied with the revision.